# Recent Advances in Molecular Research on Hydrogen Sulfide (H_2_S) Role in Diabetes Mellitus (DM)—A Systematic Review

**DOI:** 10.3390/ijms23126720

**Published:** 2022-06-16

**Authors:** Constantin Munteanu, Mariana Rotariu, Marius Turnea, Gabriela Dogaru, Cristina Popescu, Aura Spînu, Ioana Andone, Ruxandra Postoiu, Elena Valentina Ionescu, Carmen Oprea, Irina Albadi, Gelu Onose

**Affiliations:** 1Faculty of Medical Bioengineering, University of Medicine and Pharmacy “Grigore T. Popa” Iași, 700115 Iași, Romania; mariana.rotariu@umfiasi.ro (M.R.); marius.turnea@umfiasi.ro (M.T.); 2Teaching Emergency Hospital “Bagdasar-Arseni”, 041915 Bucharest, Romania; cristina_popescu_recuperare@yahoo.com (C.P.); aura.spinu@umfcd.ro (A.S.); ioana.andone@umfcd.ro (I.A.); postoiu.ruxandra@yahoo.ro (R.P.); 3Clinical Rehabilitation Hospital, 400066 Cluj-Napoca, Romania; gabriela.dogaru@umfcluj.ro; 4Faculty of Medicine, “Iuliu Hatieganu” University of Medicine and Pharmacy, 400347 Cluj-Napoca, Romania; 5Faculty of Medicine, University of Medicine and Pharmacy “Carol Davila”, 050474 Bucharest, Romania; 6Faculty of Medicine, Ovidius University of Constanta, 900527 Constanta, Romania; elena.ionescu@365.univ-ovidius.ro (E.V.I.); carmen.oprea@365.univ-ovidius.ro (C.O.); irina.albadi@yahoo.com (I.A.); 7Balneal and Rehabilitation Sanatorium of Techirghiol, 906100 Techirghiol, Romania; 8Teaching Emergency County Hospital “Sf. Apostol Andrei” Constanta, 900591 Constanta, Romania

**Keywords:** hydrogen sulfide (H_2_S), Diabetes Mellitus (DM), DM vascular-linked pathology, systematic review, oxidative phosphorylation, ROS (Reactive Oxygen Species)

## Abstract

Abundant experimental data suggest that hydrogen sulfide (H_2_S) is related to the pathophysiology of Diabetes Mellitus (DM). Multiple molecular mechanisms, including receptors, membrane ion channels, signalingmolecules, enzymes, and transcription factors, are known to be responsible for the H_2_S biological actions; however, H_2_S is not fully documented as a gaseous signaling molecule interfering with DM and vascular-linked pathology. In recent decades, multiple approaches regarding therapeutic exploitation of H_2_S have been identified, either based on H_2_S exogenous apport or on its modulated endogenous biosynthesis. This paper aims to synthesize and systematize, as comprehensively as possible, the recent literature-related data regarding the therapeutic/rehabilitative role of H_2_S in DM. This review was conducted following the “Preferred reporting items for systematic reviews and meta-analyses” (PRISMA) methodology, interrogating five international medically renowned databases by specific keyword combinations/“syntaxes” used contextually, over the last five years (2017–2021). The respective search/filtered and selection methodology we applied has identified, in the first step, 212 articles. After deploying the next specific quest steps, 51 unique published papers qualified for minute analysis resulted. To these bibliographic resources obtained through the PRISMA methodology, in order to have the best available information coverage, we added 86 papers that were freely found by a direct internet search. Finally, we selected for a connected meta-analysis eight relevant reports that included 1237 human subjects elicited from clinical trial registration platforms. Numerous H_2_S releasing/stimulating compounds have been produced, some being used in experimental models. However, very few of them were further advanced in clinical studies, indicating that the development of H_2_S as a therapeutic agent is still at the beginning.

## 1. Introduction

Diabetes Mellitus (DM) is a non-communicable chronic metabolic disease [1] characterized by prolonged hyperglycemia. Type 1 DM is a chronic condition in which the body’s pancreatic β cells, determined by different causes, reduce insulin production. Instead, type 2 DM is caused mainly by lifestyle factors and is characterized by insulin resistance, thus inefficiency on target cells, also with complex pathophysiologic links. The International Diabetes Federation mentions more than 450 million people globally with DM at present, 90% representing Type 2 DM, and estimates an increase to over 600 million in the next 25 years [2]. Therefore, the diabetic “pandemic” seems to be a global public health problem that needs particular attention, considering the deaths and disabilities caused by this disease’s complications [3].

Severe and more frequently associated complications of DM may primarily involve the nervous system [4], kidneys [5], and/or eyes [6] due to damage of the microcirculation [7], respectively; and the cardiovascular systems [8], due to macroangiopathy [9]. Cardiomyopathy is responsible for more than 50% of deaths in diabetic patients [10]. Diabetic nephropathy [11] generates kidney injuries and tissue lesions and eventually leads to chronic kidney disease.

In DM, patients with insufficient oxygen supplies due to impaired angiogenesis/neovascularization [12], refractory wounds and critical ischemic sufferance in limbs are major vascular hazards [11], connected with a malfunction of endothelial cells (ECs) [13].

H_2_S, recognized by its smell of rotten eggs, is the simplest thiol (R-SH), a sulfur analog of alcohol, with a high redox potential [14]. However, disentangling H_2_S chemistry and biochemistry is much more complicated; H_2_S biological actions are dependent on its chemical (reductive and nucleophilic) attributes. H_2_S is fast-dissolving in aqueous solutions (due to its fine acid quality), splitting to generate two anionic parts: sulfide (S_2_^−^) and hydrosulfide (HS^−^). About 81.5% of the total H_2_S exists as HS^−^ and S_2_^−^, and only 18.5% as an undissociated acid at a specific physiological potential of hydrogen (pH) of 7.4 in an aqueous solution, and thus has biological functions similar to other ionic species [3]. As a result, HS^−^ is an excellent substrate for the Anion Exchanger 1 (AE1) [15]. The acid H_2_S/HS^−^ balance follows the same principle as CO_2_/HCO_3_^−^ in the Jacobs–Stewart cycle [16]. H_2_S is highly lipophilic—a feature that allows it to freely penetrate all cell membranes without the facilitation of membrane channels, thereby exerting biological activities [17]. Additionally, it has a very high permeability coefficient in human erythrocytes. Related to this capability, it was established that in cases characterized by a lack of extracellular hydration and intracellular dehydration, the pH buffer system of the Cl−/HS−/H_2_S is faster than the Cl^−^/HCO_3_^−^/H_2_CO_3_ cycle [18]; however, in severe DM, such a pH buffer, it is much lower than that in healthy persons, and therefore, its protective effect is diminished [19].

H_2_S serves as a gasotransmitter in regulating organ development and maintaining homeostasis. Therefore, its abnormal levels are linked with multiple human diseases, such as DM, neurodegenerative diseases, myocardial injuries [20], or ophthalmic pathology [21]. Plasmatic H_2_S levels are remarkably lower in diabetic patients [3]. In addition, the levels of H_2_S in plasma, urine, and heart tissues are prone to be lower in aging diabetic rodents [22].

The striking lack of uniform data concerning H_2_S levels under physiological conditions may contribute to uncertainty about the precise mechanistic roles of H_2_S in various physiological processes. Hence, it is essential to improve the specificity of detection and reduce the threshold limits of related techniques, which may be helpful in achieving more accurate information regarding the physiological distribution of H_2_S in the blood, cells, and tissues, as well as more consistent knowledge on its typical tasks at an intimate level based on more comprehensive data on its oxidative phosphorylation activities [23] to satisfy cellular energy requirements, and perhaps other biological involvements [3].

## 2. Materials and Methods

This systematic literature review is based on the PRISMA methodology, by searching free full-text available papers written in English, which have appeared in the last five years, by specific keywords combinations (Table 1), in the well known international databases: National Center for Biotechnology Information (NCBI)/PubMed, PubMed Central (PMC), Elsevier, and Web of Science. Cochrane and PEDro databases returned no results.

The scientific impact of each article was established using a customized quantification formula to obtain a PEDro score. We considered eligible the works that received a score of at least 4 (“fair quality = PEDro score 4–5”).

To evaluate the impact of hydrogen sulfide therapeutic interventions in DM, we searched on https://clinicaltrials.gov, https://trialsearch.who.int/, and https://www.clinicaltrialsregister.eu, (accessed on 1 January 2022), for clinical trials using as items Diabetes Mellitus and H_2_S or hydrogen sulfide. The inclusion criteria were fixed regarding patients with Diabetes, age: 18 to the elderly, of all genders. Exclusion criteria correspond to study dates: not before 2017. In addition, a meta-analysis was included to analyze the different pathologies associated with diabetes and the number of subjects.

## 3. Results

### 3.1. Search and Filtering Results

Databases interrogation provided, initially, 212 articles. Appling the PRISMA selection filters and scoring resulted in 51 unique published qualified studies. We added another 94 free papers found based on Google Search by a direct internet search (Figure 1), which were highly related to our research (Table 2).

Our search on clinical trial registration platforms showed that despite the vast number of trials (25,969) on DM, only 46 included H_2_S/hydrogen sulfide. In addition, 12 were duplicates, and 26 were performed/ended before 2017. Therefore, we selected/filtred for our meta-analysis eight relevant reports that included 1237 subjects. 

### 3.2. Physiological Properties of H_2_S 

H_2_S influences many cellular processes (Figure 2) through a broad spectrum of signaling molecules, reacting with superoxide anions, hypochlorite, hydrogen peroxide, peroxynitrite, metals, thiol derivatives, and NO [60]. Moreover, with its aforementioned high rate of the anionic chemical state in aqueous solution, there are reported antioxidant properties of H_2_S that can mitigate oxidative stress-induced dysfunctions. It also acts through the potassium (KATP/K^+^) and calcium (Ca^2+^) ion channels to increase (antioxidant) glutathione (GSH) levels. GSH, Gpx (glutathione peroxidase), and superoxide dismutase (SOD) neutralize H_2_O_2_-induced oxidative damage in mitochondria. To be specified that ROS (Reactive Oxygen Species) are formed within the oxidative phosphorylation process (and in excessive quantities in such a process’ inefficiency, leading to oxidative stress and affecting mitochondrial metabolism) [11], and attenuation of mitochondrial ROS release results in completely preserved insulin sensitivity despite a high-fat diet [24].

H_2_S is also endogenously produced, like nitric oxide (NO) and carbon monoxide (CO), which are similar gasotransmitters. H_2_S has been experimentally shown to be involved in the bio-molecular regulation of vital physiological processes such as the inflammatory response, apoptosis, oxidative stress, and angiogenesis. The brain, liver, kidney, and other organs produce H_2_S [7]. The cellular biogenesis of H_2_S is based on the desulfuration of cysteine or homocysteine, a process involving mainly three enzymes: cystathionin-β-synthase (CBS), cystathionin-γ-lyase (CSE), and 3-mercaptopyruvate sulfurtransferase (MST) [61]. H_2_S biogenesis at the mitochondria level implies cysteine aminotransferase (CAT) that catalyzes L-cysteine and glutamate to 3-mercaptopyruvate and α-ketoglutarate. Furthermore, 3-mercaptopyruvate is metabolized to pyruvate and H_2_S via 3-mercaptopyruvate sulfurtransferase (3-MST) [24].

MicroRNAs are factors involved in the upregulation of CSE expression. It was also found that some currently used drugs, including angiotensin-converting enzyme (ACE) inhibitors [62], statins [14], calcium channel antagonists, aspirin, and metformin vitamin D3 [42], and many others, may increase the biogenesis of H_2_S. From this list, statins, for example, can increase H_2_S synthesis via Akt-mediated control of CSE or suppress H_2_S degradation by decreasing coenzyme Q level, a sulfide quinone reductase cofactor [57].

Several routes could eliminate the H_2_S. Firstly, H_2_S can be transformed into thiosulfate by mitochondrial oxidative modification, or further converted into sulfite and sulfate. Next, cytosolic methylation is another pathway used to transform H_2_S to dimethylsulfide by thiol S-methyltransferase. Finally, the excessive H_2_S could be scavenged by Metallo- or disulfide-containing molecules or glutathione disulfide and could also be released by the lungs [63].

Exogenously supplied or endogenously generated, H_2_S can be stored at the cellular level as bound sulfane, a reductant labile sulfur (e.g., persulfide, polysulfide, and protein-associated sulfur, among others) [14]. Human erythrocytes are about ~5 billion per mL of blood, and each has over 270 million hemoglobin molecules that can uptake H_2_S, effectively controlling its clearance. This distribution ensures the maintenance of the physiological plasma and tissue concentration of free H_2_S in the range of 15 to 150 nM. In addition, the high lipid and water solubility of H_2_S allow quick passage through the alveolar membrane, which assures an equilibrium between blood and the alveolar air level of H_2_S [18].

The potential of H_2_S metabolite products as biomarkers is appreciated since the plasmatic and urinary levels of H_2_S may reflect renal disease severity, such as chronic kidney disease [25]. Therefore, excessive exposure to H_2_S can lead to cellular toxicity, orchestrate pathological processes, and increase the risk of various diseases [64]. H_2_S is one of the most toxic poisons, and is even more harmful than cyanide on a mole-to-mole basis. A solution of dissolved H_2_S diminishes the activity of mitochondrial cytochrome c oxidase at a concentration ranging from 10 to 30 μM. In vivo studies have shown that in rodents and large mammals, severe depression of the medullary respiratory neurons and/or cardiac contractility by infusion or inhaling H_2_S at concentrations yield plasma concentrations of gaseous H_2_S between 2 and 5 μM.

H_2_S can attenuate matrix deposition and myocardial fibrosis [26] and improve MMP/TIMP disorder. The mechanism of H_2_S protection against diabetic myocardial fibrosis depends on the down-regulation of JAK/STAT and TGF-β1 (transforming growth factor) signaling [27].

Many physiological and pathophysiological properties regarding antioxidation, apoptosis, or inflammation of H_2_S are mediated through transcription factors such as Nrf2 (nuclear factor-E2-related factor), FoxO3 (Forkhead box O), and NF-kB (Nuclear factor kappa-light-chain-enhancer of activated B cells) [22]. The epigenetic role of H_2_S is unveiled by Brg1 (Brahma-related gene 1) expression modulation at the promoter region, decreasing the ATP-dependent chromatin remodeling complex’s transcriptional level, which inhibits vascular smooth muscle cell proliferation. Moreover, H_2_S may reduce the lysine acetylation of enzymes involved in fatty acid β-oxidation and glucose oxidation in diabetic statuses [28], exerting a beneficial effect on cardiac energy substrate utilization [65].

H_2_S is known to regulate various physiological functions, such as decreasing blood pressure, acting on various targets, including ion channels, such as ATP-sensitive potassium channels (KATP) [66], voltage-gated potassium channels (Kv7) [67], transient receptor potential channels (TRPV) [29], or L/T-type Ca^2+^ channels [68], mitoKATP/Kv7 channels [69]. By activating ATP-sensitive K^+^ channels, H_2_S lowers blood pressure, protects the heart from ischemia and reperfusion injury, inhibits insulin secretion in pancreatic β cells, and exerts anti-apoptotic, anti-inflammatory, and anti-nociceptive effects [70]. KATP channels also play a crucial role in insulin secretion in pancreatic cells, where the opening of the channels by H_2_S decreases insulin secretion. Both endogenous and exogenous H_2_S inhibits insulin secretion from cells by activating KATP channels and inhibiting L-type voltage-dependent calcium channels. In addition, by inhibiting glucose transporter-4 (GLUT-4), H_2_S inhibits insulin-stimulated glucose uptake in adipocytes, indicating that H_2_S decreases the insulin sensitivity of adipocytes [71].

During hyperglycemia, elevated levels of H_2_S can open the KATP channels in the islets cell membrane, which can cause high hyperpolarization and lower insulin secretion. This effect is caused by several biochemical processes that inhibit insulin secretion [69].

The endoplasmic reticulum (ER) is the cytoplasmic location where proteins are synthesized. It maintains Ca^2+^ homeostasis and participates in protein folding [72]. The molecular markers of stress include C/EBP homologous protein, cleaved caspase-12, and the glucose-controlled protein 78 (GRP78). It has been observed that chronic ER stress can trigger DM, Alzheimer’s disease, and other neurodegenerative disorders [73], engaging ER stress-induced apoptosis [10].

Studies on diabetic cardiomyopathy have shown that the effects of H_2_S on the endoplasmic reticulum stress are related to its reduction in levels of mitochondria apoptotic proteins [30,31]. The endoplasmic reticulum’s interaction with mitochondria is regulated by the ROS pathway [74]. Mitofusin-2 is a critical protein that can bridge the endoplasmic reticulum and mitochondria. It plays a role in the fusion and fission of mitochondria. It is believed that Mfn-2 is involved in the cardiac system’s mitochondria function and is triggered by oxidative stress [30]. The high levels of Mfn-2 can also induce cardiomyocyte apoptosis [32].

The mitochondria control energy homeostasis and regulate ROS production [75]. Mitochondria play a significant role in the mechanism of fatty acids β-oxidation. On the other hand, mitochondrial dysregulations occur in insulin resistance. The number of mitochondria in hepatocytes decreases in CSE-deficit cells. Hyperglycaemia leads to the generation of mitochondria superoxide, which causes the synthesis of oxidants and endothelial dysfunction. H_2_S works as an electron donor to the respiratory chain and plays a therapeutic role in DM and associated vascular diseases.

Mitochondrial DNA (mtDNA) content levels are significantly reduced in CSE-gene knockout mice. This depletion can be reversed by exogenous H_2_S gain [76]. H_2_S can provoke mtDNA replication and mitochondrial biogenesis by suppressing mitochondrial transcription factor A (TFAM) methylation. In contrast, H_2_S may stimulate cardiac mitochondrial biogenesis by activating the AMPK (5′ AMP-activated protein kinase) [33] PGC1α (peroxisome proliferator-activated receptor gamma coactivator 1-alpha) pathway [77]. Sulfhydration of AMPK and PP2A (protein phosphatase 2A) [78], which leads to AMPK activation and PP2A inhibition, respectively, has been proposed as a mechanism that may be involved in H_2_S-mediated stimulation of unstressed mitochondrial biogenesis [14].

H_2_S is a gasotransmitter with discovered roles in cellular signaling, which can also be stored as bound endosulfan, known to play a variety of physiological functions [79]. Some of these include vasodilation [80], anti-apoptosis [81], anti-inflammation [82], cell survival/death [15], cell proliferation/hypertrophy [83], endoplasmic reticulum stress [84], antioxidative stress [32], mitochondrial bioenergetics/biogenesis [50], blood pressure reduction [85], and cell differentiation [86]. H_2_S ameliorates diabetic complications, including endothelial dysfunction [87], nephropathy [34], retinopathy [88], and cardiovascular diseases.

H_2_S could also increase the apoptosis of islet cells and inhibit the programmed cell death of pancreatic cells by blocking the ERK (extracellular signal-regulated) protein kinase [33]. It has also been shown to inhibit the anti-inflammatory or antioxidant signaling pathways of pancreatic cells. Injecting H_2_S into STZ-induced diabetic rats can improve the status of their diabetes by blocking the PKC/ERK½ signaling pathway. The effects of blocking the JAK/STAT signaling pathway are also linked to the H_2_S’ anti-apoptotic effects [58]. The myocardial expressions of pro-fibrotic factors, such as MMP-2 (matrix metalloprotease 2), TIMP-2 (tissue inhibitor of metalloproteinase 2), transforming growth factor (TGF)-β1/SMAD family member 3 (Smad3) signaling pathway, and collagens are strikingly changed in diabetic rats. Many studies have shown that suppressing the STAT3 pathway can improve the physiological effects of H_2_S. It can also contribute to the cardioprotective effects of H_2_S by reducing the levels of ROS in the body [89].

H_2_S can also activate the soluble guanyl cyclase (sGC) by directing its interaction with the cGMP phosphodiesterase (PDE). In addition, this molecule can trigger the activation of the cyclic GMP-protein kinase G pathway [90]. It can also trigger the re-translation of eIF2 (eukaryotic initiation factor 2) [91] by increasing the phosphorylation of protein phosphatase-1 [27].

H_2_S can also convert the -SH group of cysteine into a -SSH group, which can alter the activities of various enzymes such as the F_1_F_0_-ATPase pump, KATP channels, and the phosphatase and tensin homolog (PTEN) [14]. This can lead to the disappearance of certain protein S-sulfates. S-sulfhydration is a post-translational process that produces a hydropersulfide moiety or polysulfide in specific body regions. It is known to regulate the cellular functions of H_2_S [92]. HMG-CoA reductase [35] is an enzyme involved in the ubiquitination of various substrate proteins, such as Hrd1. H_2_S induces the degradation of VAMP3 (vesicle-associated membrane protein 3), which controls exocytosis by Hrd1 S-sulfhidration. It is also known to trigger the translocation of CD36, which can cause lipid toxicity in the body [58].

Cytokines are small molecules that help the cell produce pro-inflammatory signals. Increased production of cytokines in the serum and heart muscle is a common feature of cardiovascular disease involving cell death. The JAK/STAT signaling pathway is a vital pathway for cytokine signal transduction and a pleiotropic cascade involved in growth hormone receptors’ activity and regulates various physiological and pathological processes, including proliferation, differentiation, apoptosis, and cellular immunity, and inflammation. In addition, this pathway can also increase the expression of TGF and type III collagen [93].

The pro-inflammatory cytokine TNFα can also induce apoptosis and necrosis. The stimulation of TNFα in the liver and macrophages can increase the secretion of H_2_S. It has been observed that the treatment with LPS leads to an increase in the production of both IL-6 and TNFα, an epigenetic regulation mechanism [94].

In addition, treating patients with H_2_S can decrease the production of neutrophils in the myocardium and contribute to the development of anti-apoptotic signaling. Neutrophils are recruited into the myocardium to express IL-1β and TNFα. H_2_S reduces these immunity cells, correlated with promoted Bcl-2 anti-apoptotic signaling, decreases cytokine release, and preserves cardiac function [95].

VEGF is a pro-angiogenic cytokine that promotes endothelial cell survival. In the case of acute coronary syndrome, the reduction of VEGF leads to the depletion of microvessels. On the other hand, H_2_S can prevent coronary artery disease and improve the survival of endothelial cells [96].

TGF is a critical cytokine in the development of cardiac remodeling. Myofibroblasts can promote the growth and deposition of collagen in the body. However, the presence of H_2_S can inhibit the signaling cascade in myofibroblasts and limit the proliferation and survival of these cells. The cytoprotective effects of H_2_S on cell death appear to act through cell types other than cardiomyocytes, where it influences TGFβ expression and inhibits the signaling cascade in fibroblasts. This restricts the differentiation and proliferation of fibroblasts into myofibroblasts and prevents the deposition of collagen in the heart [22].

H_2_S and NO are physiological and pathological factors that have been extensively studied lately. They have been linked to the development of diabetes and heart failure [92]. The exposure of mice to H_2_S can stimulate the production of NO through the activation of the eNOS pathway [97]. This can result in the development of more severe cardiac dilatation. Treating patients with H_2_S using CSE overexpression can also improve the function and structure of their hearts after undergoing transverse aortic contraction. This therapy activates the eNOS-NO-cGMP pathway [36] and can prevent hepatic and myocardial ischemia-reperfusion injury [98,99,100].

### 3.3. H_2_S in Pharmacology and Pathophysiology

H_2_S levels are decreased in several conditions (e.g., DM, ischemia, and aging), even in COVID-19 [37], and are elevated in other statuses (e.g., inflammation, critical biological disbalances, and cancer). In recent decades, multiple approaches to the therapeutic exploitation of H_2_S have been identified, either based on H_2_S exogenous apport or decreased H_2_S biosynthesis [43]. Inhibition and stimulation of H_2_S synthesis have been suggested as potential interventions in DM. 

Treating patients with H_2_S can improve the recovery of liver and myocardial ischemia-reperfusion injury. It can also restore the damaged endothelium-dependent relaxation caused by NO depletion [13]. H_2_S is known to produce nitroxyl, a one-electron reduction of NO. It can also help restore the relaxation caused by the depletion of NO [101]. 

Various studies have also shown that H_2_S can promote the development of new blood vessels. Most of these studies were focused on the effects of VEGF on the angiogenic response [11]. Silencing of CSE by siRNA can also decrease the impact of VEGF-induced angiogenesis. It can also stimulate the activity of various cellular signaling pathways, such as the eNOS-NO-c pathway and the K1A2T7P signal transducer. H_2_S promotes angiogenesis by increasing the activity of endothelial nitric oxide synthesis (eNOS), phosphatidylinositol 3 (PI3)-kinase/protein kinase B (AKT), p38/MAPK, K1A2T7P, signal transducer and activator of transcription 3 (STAT3), and sirtuin 1 (SIRT1)/VEGF/cyclic guanosine 5′-monophosphate (cGMP) cascade [102].

Although diabetes has been known to impair the development of new blood vessels [13], the mechanism of this process is not yet precise. The absence of vascular perfusion leads to diabetes-induced angiogenesis. H_2_S rescues the migration of HUVECs in mice with hyperglycemia-induced migration. The effects of this condition on the pro-angiogenic and bio-energetic properties were also studied. H_2_S improves the revascularization of diabetic mice through increasing NO bioavailability and promotes the development of vascular progenitors [86].

Under pathological conditions, the levels of H_2_S and its production enzymes are significantly altered [103]. This can lead to the development of various cardiac disorders [38]. In addition, H_2_S increases the filtration rate and kidney blood flow [62] and generates an increase in the excretion of certain nutrients, such as K^+^ and Na^+^. The role of the Renal-Aangiotensin system (RAS) [22] is well established in the pathogenesis of various diseases. It plays a central role in regulating physiological function and possesses neuronal control of the circulatory system. H_2_S is also known to interact with the zinc metalloproteinase, a zinc metalloproteinase. In addition, studies show that this protein can reduce the activity of the angiotensin-converting enzyme (ACE) in human endothelial cells.

Myocyte stretching releases angiotensin II (ANG II) [3], increases p53 binding to the ANG II promoter and the AT1 (angiotensin II type 1) receptor, and results in a four- to seven-fold increase in apoptosis. Adding Zn^2+^ to the diet lowered the ACE mRNA level and reduced ROS production. H_2_S could also alter RAS signaling, interacting with the ACE, a zinc metalloproteinase [62]. A dose-dependent drop in ACE activity in human endothelial cells after treatment with H_2_S was observed. Supplementation of H_2_S in DM rats reversed RAS activation and reduced ROS production. H_2_S could alter RAS signaling, reducing oxidative stress [22].

As a neuromodulator, H_2_S can improve the effects of diabetes on the central nervous system (CNS). Due to the impact of diabetes on the CNS, it is considered a leading cause of cognitive decline [39]. H_2_S can also reduce the risk of cognitive decline and microvascular complications [94]. An equilibrated balance of oxidative stress/antioxidants is essential for maintaining cellular function. When this is disturbed, the other molecules, such as deoxyribonucleic acid, lipid, and protein oxidize, imprinting a pathological condition, like diabetes [24]. Oxidative stress comes from the overproduction of reactive oxygen and nitrogen species. The main source of ROS is the mitochondria [99]. Oxidative stress in diabetic patients determines dysfunctions during insulin secretion in the nervous system, and thus, neurodegeneration, such as diabetic peripheral neuropathy (DPN), occurs [104].

The pancreatic β cell is the most essential metabolically active part of the body, where metabolites take place for energy synthesis at the high glucose concentration level. H_2_S displays antioxidant effects by directly silencing reactive oxygen species (ROS) via a hydrosulfide anion (HS-), a powerful one-electron chemical reductant dissociated from H_2_S in a physiological fluid. H_2_S can improve the function of the mitochondria, which is a type of respiratory chain that produces oxygen [105]. Overproduction of reactive nitrogen and oxygen species can lead to oxidative stress. The free radicals produced by these species can be suppressed by antioxidant molecules [14]. H_2_S can also decrease ROS production by suppressing the copper/zinc superoxide activation. In addition, it can also prevent the degradation of antioxidant enzymes and proteins [63].

Autophagy is emerging as a critical cellular stress response that is involved in a variety of disease states. Autophagy is a highly conserved self-feeding pathway that degrades macromolecules and damaged organelles to maintain intracellular homeostasis. It has been shown that H_2_S is a regulator of autophagy. Generally, autophagy serves a dual purpose: it may play a cytoprotective or harmful role in the body, hanging on the type and severity of the lesion it causes [22]. A certain degree of autophagic activity is essential in promoting tissue homeostasis and cell survival. However, excessive autophagic activity can contribute to apoptosis on the other side. In addition, autophagy dysfunction is involved in diabetic cardiomyopathy [106].

Autophagy is a well-coordinated, multi-stage process regulated by autophagy-related genetic products and proteins, such as Beclin1 and P62. Exogenous H_2_S facilitates the elimination of autophagosome contents, which improves autophagy. The promotional effects of exogenous H_2_S on autophagy may be essential for decreased ROS production. In addition, there are studies that ubiquitin aggregate clearance is mainly dependent on autophagy, and disruption of autophagy results in the accumulation of ubiquitin aggregates in cells [33].

H_2_S has its regulatory role in autophagy during the development and progression of numerous diseases, such as diabetes, heart failure, or Parkinson’s disease [16]. Exogenous H_2_S reduces the ubiquitination level. Recent studies have found that Keap-1 is crucial in eliminating ubiquitin proteins [107]. Keap-1 can be a critical factor in the protective role of exogenous H_2_S on ubiquitin aggregate clearance via autophagy [16]. Exogenous H_2_S upregulates the expression of Keap-1. Reported data show that Keap-1 regulates the translocation of Nrf2, a negative regulator of ROS production. However, exogenous H_2_S had no significant effects on the translocation of Nrf2 to the nucleus. A recent study demonstrated that H_2_S suppressed diabetes-accelerated atherosclerosis via Nrf2 [108].

H_2_S rectifies high glucose/palmitate-induced excessive autophagy in endothelial cells. The Nrf2-ROS signaling pathway can trigger this effect. However, exogenous H_2_S inhibits mitochondrial apoptosis and promotes mitochondrial autophagy, thus protecting endothelial cells against apoptosis induced by high glucose and palmitate. Therefore, it has been hypothesized that H_2_S can promote the normal development of the diabetic endothelial system by suppressing the excessive autophagy that occurs following stressful events [44]. The optimal window of autophagy is maintained in response to stressful events. However, if it is excessive, autophagy is maladaptive, leading to cell death [22].

Some studies showed that H_2_S upregulates autophagy and others that H_2_S inhibits autophagy. H_2_S plays diverse roles in autophagy depending on the tissue and disease. For example, H_2_S could downregulate LC3BII and Beclin-1 protein expression and upregulate p62 protein expression in VSMCs (vascular smooth muscle cells) under HG (high glucose) conditions, which could be reversed by rapamycin, an autophagy activator. Furthermore, NaHS decreased the autophagy induced by HG in VSMCs. Similarly, ALA (Alpha-lipoic acid) could also inhibit autophagy in VSMCs under HG conditions via the AMPK/mTOR signaling pathway. Autophagy is regulated by many signaling pathways, among them the AMPK/mTOR signal pathway being crucial. The activation of the AMPK/mTOR pathway in DM has been widely studied. Increased AMPK phosphorylation and decreased mTOR phosphorylation activate autophagy [33]. H_2_S also downregulates autophagy via the AMPK/mTOR signaling pathway [33].

An essential regulator of inflammation associated with metabolic syndrome is the nucleotide-binding domain, leucine-rich-containing family, pyrin domain containing-3 (NLRP3) inflammasome, which activates caspase-1, after interacting with the adaptor protein apoptosis-associated speck-like protein containing a C-terminal caspase recruitment domain (ASC). Induction of phosphorylation of the p65 subunit of NF-κB resulting in NF-κB signaling activation is a prerequisite for transcriptional activation of NLPR3 [109]. Cleavage, processing, and secretion of pro-inflammatory cytokines IL-1β and IL-18 result from NF-κB-mediated activation of NLRP3 inflammasome and subsequent caspase-1 activation [45].

H_2_S can exert anti-inflammatory effects against free fatty acid (FFA)-induced inflammation and apoptosis in macrophages by suppressing TLR4/NF-κB-stimulated NLRP3 inflammasome activation. H_2_S can thus prevent FFA-overload-mediated insulin resistance and type 2 DM [110].

H_2_S exerted an anti-inflammatory role in diabetic myocardia by downregulation of Thioredoxin-interacting protein (TXNIP)-mediated NLRP3 inflammasome activation. H_2_S alleviated hyperglycemia-mediated myocardial inflammation in type 1 DM. The mechanism may involve inhibiting TXNIP-mediated NLRP3 inflammasome activation, which might serve as an efficient, targeted therapy in diabetic cardiomyocytes [46].

Pyroptosis is a type of cell death with several characteristics that make it different from other forms of cell death. This type of cell death relies on the canonical pathway, dependent on caspase-1 [47], and the non-canonical pathway, reliant on caspase-11 [111].

Since pyroptosis is known to trigger the inflammatory response that contributes to chronic inflammatory diseases, it has shifted its focus away from the body’s natural defenses. The downstream inflammatory markers of pyroptosis are associated with toxic shock, nephropathy, and pathogen defense. The NLRP3 (NOD-like receptor protein 3) inflammasome activates caspase-1 in the canonical pyroptosis pathway. NLRP3 also localizes to the mitochondria and supplies high ROS production, but it can be inhibited by H_2_S [48].

The Mfn-2 protein is known to promote early apoptotic events in the mitochondria. The fragmentation of the mitochondria network can also lead to the development of these events. It has also been shown that Mfn-2 can prevent the transfer of Ca^2+^ from the endoplasmic reticulum to the adjacent mitochondria. High glucose levels can also promote the growth of H9C2 cells through the increased expression of Mfn-2- and siRNA-mediated Mfn-2 silencing [30,49].

Necrosis is a version of cell death that occurs following severe injury. It is programmed to utilize the TNF receptor and the RIPK1/RIPK3 necrosome. The effects of H_2_S on necroptosis and necrosis are limited. However, recent studies have shown that treating cardiomyocytes with high glucose levels can inhibit these markers [112].

A non-canonical death pathway known as MPT (mitochondrial permeability transition pore) is also initiated by Ca^2+^ and ROS. These stressors open the nonspecific MPT pore in the mitochondrial inner membrane, dissipate inner membrane potential, and rupture both mitochondrial membranes through osmotic swelling. The absence of the outer membrane can prevent the formation of apoptotic bodies. Instead, cell death occurs through the accumulation of necrosis. Data show that H_2_S protects against MPT-driven necrosis in the heart and brain [113].

### 3.4. H_2_S and Insulin Secretion and Sensitivity

Insulin resistance and compromised insulin secretion lead to impaired glucose metabolism, which contributes to the development of Diabetes [17]. H_2_S could be produced endogenously in the pancreatic island’s β cells, liver, fat, skeletal muscle, and hypothalamus and regulates local and systemic carbohydrates metabolism [51]. Specifically, H_2_S is reported to suppress insulin secretion and promote or reduce islet β-cell apoptosis. It influences insulin sensitivity. H_2_S also suppresses glucose uptake and glycogen storage and promotes or inhibits gluconeogenesis, mitochondrial bioenergetics [50], and mitochondrial biogenesis in the liver [52]. This gas also promotes glucose uptake into adipocytes in the fat tissue, while other studies have reported inhibiting this process. H_2_S has been shown, as well, to increase adipogenesis, inhibit lipolysis, and regulate adiponectin and MCP-1 secretion in adipocytes [53]. H_2_S increases glucose absorption in skeletal muscle, improves insulin sensitivity and modulates circadian clock genes in myocytes. The hypothalamic CBS (cystathionin-β-synthase)/H_2_S pathway reduces obesity [54] and improves insulin sensitivity through brain–adipose interactions. Most studies have shown that plasmatic H_2_S levels are lower in diabetic patients [50,55].

H_2_S can influence insulin secretion and modulate circulating glucose levels. H_2_S administration to β cell lines attenuates insulin secretion triggered by a high glucose concentration. High levels of H_2_S can also decrease the secretion of insulin. It can also cause the membrane to become polarized and inhibit the KATP channel’s independent signaling. H_2_S can also inhibit insulin secretion by affecting various biochemical processes: activation of KATP channels, inhibition of ATP synthesis, and inactivation of L-type voltage-dependent Ca^2+^ channels [114].

The effects of hyperglycemia on insulin secretion can vary depending on the phase of diabetes development. During the early stages of the disease, increasing levels of H_2_S can protect islet cells from further damage. During the development of diabetes, an increase in H_2_S can inhibit the secretion of insulin and reduce the overload of islet cells. This can also trigger an increase in ER stress response [50].

Inhibitory effects of sodium hydrosulfide (NaSH, 10 μM^–1^ mM) and L-cysteine (0.1–10 mM) on glucose (10 mM)-induced insulin secretion has been observed in both isolated mouse islets and pancreatic β cell lines, an effect that was not observed at a low glucose concentration (3 mM) [115]. One of the mechanisms through which H_2_S inhibits insulin secretion is through the opening of KATP channels, as the inhibitory effects of NaSH and L-cysteine on insulin secretion were reproduced after using tolbutamide (a KATP blocker), α-ketoisocaproate (a mitochondrial fuel), and high K^+^ condition (30 mmol/L). Interactions between H_2_S with KATP channels seems to be mediated through functional manipulation, probably by decreasing selective cysteine residues of the KATP channel protein, independent of cytosolic second messengers. It has been suggested that the S-sulfhydration of KATP channels is a mechanism by which H_2_S could influence insulin secretion [56].

Hepatic insulin resistance reveals the failure of insulin to inhibit glycogenolysis and gluconeogenesis in the liver to maintain normal plasma glucose levels. The enzymes CSE, CBS, and 3-MST, responsible for endogenous H_2_S, are found in the liver. The effects of diabetes mellitus and its related pathologies on the H_2_S production system in the liver are controversial. Compared with nondiabetic rats, H_2_S production and CSE and CBS mRNA levels in the liver were increased in STZ diabetic rats, while insulin treatment reversed these effects [114]. H_2_S regulates glucose uptake, glycogen storage, and gluconeogenesis H_2_S is a key component of liver glucose metabolism [50].

### 3.5. H_2_S and Neurological Dysfunctions as Diabetes Associated Diseases

Neurological research concerning diabetes patients with complications such as Alzheimer’s disease, Parkinson’s disease, and amyotrophic lateral sclerosis present changes in the central nervous system because of the high blood glucose levels (HbA1c) correlated with poor cognitive function. Oxidative stress plays a part in inhibiting insulin signaling, which is necessary for brain function [110]. The regulation of Schwann cells, aggregation of sorbitol during signaling of the polyol pathway, inactivation of Na^+^/K^+^ signaling, and hyperglycemia-induced oxidative stress are causative factors of neuropathy in the brain. About 50% of diabetic patients are affected by neurological disorders, the most common comorbidities of DM. In addition, some neurodegenerative diseases, like Alzheimer’s disease (AD), Parkinson’s disease (PD), and amyotrophic lateral sclerosis (ALS), coincide in the central nervous system (CNS) in diabetic patients because of oxidative stress [24].

H_2_S inhibits Aβ-induced neuronal apoptosis by regulating mitochondrial function. In addition, H_2_S could inhibit the expression of IL-23/IL-17 axis and mitochondrial apoptotic proteins to alleviate the cognitive decline caused by DM [32].

The repercussions of hyperglycemia-induced oxidative stress on neurons vary in T1DM and T2DM, and DPN patients with T2DM have a low capacity to control hyperglycemia. In the peripheral nerves of T2DM patients, oxidative stress is increased from proximal to distal parts, passes from DRG to the sciatic nerve, and decreases the metabolism under the glycolytic and tricarboxylic acid cycle [29]. In AD, cholinergic homeostasis is hampered by downregulation of insulin/insulin growth factor (IGF) resistance, leading to downregulation of target genes. There is evidence that 40% of patients with DM develop PD, and glucose is impaired at an early stage. However, H_2_S can mitigate the effects of oxidative stress on nerve cells. 

### 3.6. H_2_S and Cardio-Vascular Dysfunctions as Diabetes Associated Diseases

Cardiovascular complications frequently cause hospitalization and death among diabetic patients. The early-onset diastolic dysfunction is the main characteristic of diabetes cardiomyopathy (DCM), an independent complication of diabetes, secondary to myocardial fibrosis [30]. DCM is a type of cardiomyopathy of unknown etiology, responsible for 75% of idiopathic dilated cardiomyopathy cases among diabetic patients. DCM characteristics are represented by impaired myocardial insulin signaling, mitochondrial dysfunction, overstimulation of the sympathetic nervous system, oxidative stress, increased inflammation, coronary microcirculation dysfunction, and inadequate immune response. These pathophysiological changes lead to fibrosis, hypertrophy, cardiac diastolic and/or systolic dysfunction, and ultimately, heart failure [35].

One of the alarming structural characteristics of DCM is represented by the overproduction and deposition of myocardial interstitial collagen, which leads to cardiac interstitial fibrosis, myocardial rigidity, and cardiac dysfunction. Although the precise mechanism of these changes has not been fully elucidated, the available literature suggests the cellular implication of oxidative stress, cell apoptosis, autophagy, inflammation, and endoplasmic reticulum stress are the main triggers [16].

Homocysteine transsulfuration produces H_2_S, a gaseous signaling molecule with a cardioprotective role that is capable of preventing cardiac remodeling, cell death, and pyroptosis. Research activity in vitro and on mouse models demonstrated that H_2_S inhibits caspase-1 activity and IL-1 secretion, with important suppression activity on pyroptosis in ischemic cardiomyopathy [116].

H_2_S has reducing hypertensive effects and has an important protective role in cardiomyopathy models. Moreover, H_2_S is an essential signaling molecule of the cardiovascular system with physiological and pathological mechanisms in ensuring homeostasis [80].

Accelerated atherosclerosis is a common cardiovascular complication in diabetic patients [117]. With a much higher incidence than in non-diabetic patients, atherosclerosis has an earlier onset and a higher mortality rate. Unfortunately, there is no proven treatment capable of slowing down atherosclerosis in DM. On the other hand, H_2_S has important effects on atherosclerotic plaque stabilization and on hyperglycemia-induced endothelial dysfunction, being capable of ischemia-reperfusion injury, myocardial infarction, and heart failure prevention [118].

Different pathological states of the venous and/or arterial system characterize the vascular dysfunction or vascular disease, a pathology capable of inducing adverse cardiovascular events. We mention atherosclerosis, arterial remodeling, thrombosis, and restenosis, among these pathologies. The main cardiovascular risk factors (diabetes, obesity, hypertension, aging) are responsible for vascular dysfunction through mechanisms such as oxidative stress, an essential target in therapeutic and preventive strategies. Cellular oxidation is a tightly regulated process involving both pro- and antioxidant systems from different cellular compartments in physiological conditions [87].

Results from current literature support the multiple beneficial roles of H_2_S in diabetic cardiovascular complications. First, H_2_S slows down the onset and improves the prognosis of diabetic cardiomyopathy. Second, H_2_S treatment ameliorates high-fat diet (HFD)-induced cardiac dysfunction through sulfide levels restoration; activation of adiponectin- AMPK signaling and decrease in HFD-induced ER stress secondary to H_2_S underlines its protective effects. Third, adiponectin’s essential cardiovascular protective role strengthens the correlation between low adiponectin levels and high cardiovascular risk. Fourth, adiponectin uses the AMPK to deliver its metabolic regulatory effects. AMPK increases the expression of GLUT4, which stimulates glucose transport and modulates fatty acid oxidation and cardiac lipid accumulation through the phosphorylation and inhibition of acetyl-coenzyme [50].

In recent studies, H_2_S was shown to be involved in the regulation of various vascular conditions, such as nephropathy, retinopathy, and neuropathy. H_2_S-releasing agents could potentially be used as a treatment for diabetes-related endothelial dysfunction. They could help restore the function of the vascular endothelial cells [7].

Many authors also noted that the use of H_2_S-releasing agents could be beneficial for treating diabetes by blocking the formation of advanced glycation end products (AGEs), which can lead to the development of vascular complications, and contribute to the degradation of the endothelium’s functionality. The rats that were treated with H_2_S-releasing agents exhibited a decrease in their vascular oxidation stress levels. This beneficial effect was also partially explained by the compound’s ability to increase the NO level [7].

### 3.7. H_2_S and Renal Dysfunctions as Diabetes Associated Diseases

In renal physiology, H_2_S induces vasodilation and increases renal blood flow and glomerular filtration rate, resulting in an indirect increase in the urinary excretion of Na^+^ and K^+^. In addition, H_2_S exhibits an inhibitory effect on specific Na^+^ and K^+^ kidney transporters, thus further increasing the excretion of such electrolytes into the urine. Furthermore, H_2_S acts as an oxygen sensor in the renal system, especially in the medulla. Moreover, H_2_S is found to inhibit renin release in rat models of renovascular hypertension. Hypertension-related nephropathy, a consequence of long-term hypertension, is the second leading cause of chronic kidney disease in the world. The blood pressure-lowering [119] actions of exogenous H_2_S donors have been demonstrated in spontaneously hypertensive rats, angiotensin II-induced hypertension, N^w^nitro- L-argininemethyl ester (L-NAME)-induced hypertension, and renovascular hypertension [120]. Furthermore, renal protective effects of H_2_S are observed in hypertensive animal models [121].

In Diabetic Nephropathy (DN) [122], the increased expression of TGF-β1 has been shown to promote the accumulation of ECMs such as collagens and fibronectin, apoptosis, dedifferentiation of podocytes, and epithelial-mesenchymal transition of proximal tubules, all of which are considered to facilitate renal hypertrophy and dysfunction. ERK½, a member of the MAPK family, may be expressed in mesangial cells in the condition of high glucose. ERK½ may upregulate TGF-β1 expression. Dysregulation of matrix metalloproteinases, (MMPs) or tissue inhibitors (TIMPs), are involved in the mechanism of renal fibrosis. MMPs are responsible for extracellular matrix degradation. MMPs and TIMPs construct a time-and-space-dependent system [68]. Treatment with H_2_S could attenuate the progression of renal dysfunction in diabetic rats. The protective effects of H_2_S are correlated with TGF-β1 signaling through the ERK_1/2_ pathway [47].

### 3.8. H_2_S Exogenous Sources as Possible Therapeutic Interventions in Diabetes or Related Diseases

H_2_S is, on the one hand, a therapeutic natural gas [17] that is found in mofettic joints with carbon dioxide [123], in sulfurous waters [124], with appraised medical effects in balneotherapy [4], and, on the other hand, an endogenous gaseous signal substance in the organisms. Therefore, experimental animals can be exposed to an H_2_S-rich environment to observe this gas’s physiological effects or toxicity. Reports show that when mice were exposed to 80 ppm of H_2_S for 6 h, their oxygen intake dropped by ~50%, and the metabolic rate and core body temperature were also seriously decreased into a suspended animation state. Notably, lowering metabolic demand could help reduce tissue/cellular damage caused by trauma. However, a later study of other larger species indicated that H_2_S only exerted thermoregulatory effects. In diabetes, H_2_S could promote glucose uptake by ameliorating insulin resistance and reducing renal injury [125].

Peloid or therapeutic mud is a maturated mud with healing properties, composed of a complex intermixture of fine-grained natural substances of geologic and/or biologic origins, water, and standard organic composites from biological metabolic activity. Sapropelic muds or sapropels are found at the bottom of salt waters, originating from the action of microorganisms on flora and fauna of the water basin [17]. The gaseous phase of sapropelic mud results from the biochemical processes involved in the mud formation (peloidogenesis): H_2_S, CO_2_, NH_4_, CH_4_, O_2_, and Rn. H_2_S has been reported as an active molecule of the mud, which can be absorbed through the skin [126], exerting numerous pharmacological effects. Under the action of mud, there is a harmonic stimulation in all glands to increase the enzymatic and synthetic activity, while maintaining the specificity of each. Usually, mud therapy is contraindicated in diabetic patients without glycemic control. Future research is necessary to elucidate the implications of mud therapy on diabetes [127].

Although less rigorously described in the scientific literature, H_2_S is commonly used in the context of balneotherapy, where H_2_S inhalation occurs as humans are soaking in H_2_S-containing sulfurous waters, with at least 1 mg/L of H_2_S. Hydrogen sulfide delivery into the body probably occurs via inhalation and absorption through the skin [128] or, in specific cases, when patients are sitting in closed rooms with H_2_S donors and H_2_S fountains of H_2_S-containing thermal water placed in the middle of the room, where a sensor/ventilation feedback system regulates the H_2_S concentration in the air of the room [124]. Small-scale preclinical studies demonstrate the beneficial effects of H_2_S delivery via sulfurous waters [129]. In addition, exploratory clinical studies suggest the anti-inflammatory effects of ultrasonic nebulization with sulfurous water in asthmatic patients. However, the potential therapeutic effect of these approaches has not been studied in appropriately powered, randomized clinical trials [130].

One of the potential problems with all forms of H_2_S delivery, but especially with H_2_S inhalation, relates to the issue of possible overdosing and consequent intoxication. Although the inhibitory effect of H_2_S on mitochondrial Complex IV is reversible and therefore supporting therapy can result in patient recovery in some cases, there are currently no well-characterized pharmacological antidotes to H_2_S intoxication [131].

Under physiological pH, H_2_S is in a specific equilibrium with HS- in aqueous solutions. The HS- and H_2_S are in an 81 to 19% report. Inorganic sulfide salts, such as sodium sulfide (Na_2_S) and sodium hydrosulfide (NaHS), are frequently used as H_2_S equivalents in many kinds of research [132]. These salts are fast H_2_S donors, as they produce H_2_S after being dissolved in aqueous solutions [21].

The rapid volatilization of H_2_S can cause it to escape from the buffers. This phenomenon could explain the discrepancy between the physiological responses required to trigger physiological responses in tissues and blood [133].

Many studies have used NaHS as a standard H_2_S donor. For example, it was shown that NaHS could alleviate amyloid beta-peptide (Ab)-induced neural lesion in an Alzheimer’s disease cellular model. Furthermore, in hypoxic skin damage, NaHS could exert anti-inflammatory effects through inhibition of reactive oxygen species (ROS)-activated NF-kB/cyclooxygenase (COX)-2 [134].

Allicin is commonly used as a sulfur-containing compound in garlic. It can be considered an active H_2_S pool. In aqueous solutions, it can transform various sulfur-containing combinations into H_2_S. In contrast, the diallyl disulfide (DADS) produces only a limited amount of H_2_S after a slow reaction with GSH. This process can be initiated by forming a cyclic disulfide [2].

### 3.9. Synthetic Slow-Releasing H_2_S Donors

The types of donor that can be considered controlled are those with various release mechanisms [135]. Since using H_2_S gas or sulfide salts in studies has been deemed dangerous, researchers have focused on synthetic molecules releasing H_2_S [21]. For example, GYY4137 is a Lawesson’s reagent that can be used as a slow and safe source of H2S. However, it is not as effective as an aqueous solution and can only be administered on animals. Another commonly used method is using sulfur-containing dithiolethione [132].

Thiomolybdate salts are thiol transfer reagents in organic synthesis [21]. The four sulfur atoms they present in their structures make them excellent copper chelators. Ammonium tetrathiomolybdate (TTM) can release H_2_S under strongly acidic conditions. As such, it is possible to use TTM as an inorganic complex-based H_2_S donor. TTM is a slow H_2_S releaser. It was discovered that acidic pH increase TTM’s H_2_S release [43].

In the last years, several ROS-activated H_2_S donors were designed. For instance, carbonyl sulfide can be released through a cyclic anhydrase reaction. This process can be sped up by carbonic anhydrase. The tandem reaction will remove carbonyl sulfide (COS), as well as quinone and amine byproducts [136].

Further studies reveal that donors can also release H_2_S through the intervention of an endogenous H_2_O_2_ in their cells. This method is similar to the cyclic anhydrase reaction. COS can easily undergo hydrolysis to produce H_2_S if carbonic anhydrase (CA) is presented. However, studies showed that CA is unnecessary for the donors’ H_2_S release, as H_2_O_2_ can also trigger the rapid H_2_S release from COS [21]. The effects of oxidizing stress on the cell viability of donated human tissue were studied [76]. It was revealed that these individuals exhibited the most effective outcomes of oxidizing stress on their cells [137].

Recently were also communicated a series of esterase-activated H_2_S donors. Association of esterase-activated donors with NSAID can form hybrid anti-inflammatory and anti-oxidative combined drugs that reduce NSAID-induced gastric damage [21].

Researchers also discovered nitroreductase-activated donors that could be used to release H_2_S [21]. A type of H_2_S-producing material is the polyNTA. This substance contains N-thiocarboxyanhydrides and undergoes a ring-opening reaction to donate H_2_S [21]. A PEG-ADT (5-4-hydroxyphenyl-3H-1,2-dithiole-3-thione-conjugated with polyethylene glycol) can also be used to generate H_2_S. Cell imaging studies also showed that PEG-ADT could enter cells through the endolysosome and last in the cytoplasm [21,43].

### 3.10. H_2_S-Stimulating Agents

Aside from H_2_S donors, some compounds can also stimulate the production of H_2_S in vivo. For instance, the amino group L-cysteine is an essential substrate for the enzymes that produce H_2_S. When acetylated, the resulting product N-acetyl-L-cysteine can increase the production of H_2_S [138]. Two other cysteine derivatives, S-allyl-L-cysteine and S-propargyl-L-cysteine, can be used as CSE substrates to generate H_2_S [139].

Vitamin D [11] is known to promote the growth and remodeling of bones [21]. In addition, researchers discovered that vitamin D could increase the concentration of H_2_S in the liver and kidney. Notably, it was found that cholecalciferol, known as VD3, could increase tissue H_2_S concentration in mouse heart, brain, and kidney. Meanwhile, another report suggested that VD3 could upregulate glucose transporter type 4 (GLUT4) and decrease glycemia in diabetes through stimulation of CSE expression and H_2_S generation [21,50].

### 3.11. Clinical Studies on H_2_S Donors/Exogenous Sources in Diabetes or Related Diseases—Meta-Analysis 

Presentation of clinical studies on H_2_S donors/exogenous sources in diabetes or related diseases (Figure 3, Table 3)
https://clinicaltrials.gov✓17,498 Studies found for: Diabetes Mellitus✓2 Studies found for: H_2_S/Hydrogen sulfide|Diabetes Mellitus✓18 Studies found for: hydrogen sulfidehttps://trialsearch.who.int✓6430 trials found: Diabetes Mellitus✓0 Studies found for: H_2_S/Hydrogen sulfide|Diabetes Mellitus✓25 trials found for: hydrogen sulfidehttps://www.clinicaltrialsregister.eu✓2041 trials found: Diabetes Mellitus✓0 Studies found for: H_2_S/Hydrogen sulfide|Diabetes Mellitus✓1 trial found for: hydrogen sulfide


## 4. Discussion

DM has become a significant risk factor for human health. The worldwide incidence of this disease has steadily increased due to higher rates of obesity and bad lifestyle habits. It is also associated with several fatal complications, such as cardiovascular illnesses, which account for most of the morbidity and mortality in the diabetic population. Furthermore, plasma H_2_S levels are negatively linked with HbA1c, duration of this sickness, and systolic and diastolic blood pressures [24].

It must be additionally specified that the pathogeny of DM has a common ground with the ischemia that generates a stroke: in both pathologic states, there is a depletion of ATP, the primary energy provider of metabolic processes [140], and an increase in the oxidative stress at the cellular level. Interestingly, H_2_S is indicated as a therapeutic intervention in these two pathological conditions [141].

On the other hand, H_2_S has been previously considered, inclusive in occupational medicine, as a poisonous and occasionally lethal toxic gas [66] formed from the decomposition of various organic materials. Therefore, it might also represent an industrial safety hazard, too, as it is colorless. However, H_2_S is a toxic byproduct of microbial metabolism in the atmosphere, depending on its concentration. The human nose could detect H_2_S at a level of 0.1 ppm [3].

H_2_S is a widely used reducing agent that has unique chemical properties. It can be availed to target in this purpose various cellular and molecular components due to its nucleophilic nature. In addition, it can rapidly lose its chemical identity under multiple conditions, such as a tissue bath. For instance, under aerobic conditions, the half-life of H_2_S is about 2.0 min in human hepatic cells, 2.8 min in kidney tissues, and 10.0 min in brain homogenates [27].

H_2_S is a metabolite of sulfur amino acids in mammals, aside from SO_2_ (sulfur dioxide) and Taurine. Taurine methionine, cysteine, and homocysteine are the four most common sulfur-containing amino acids, but only methionine and cysteine are incorporated into proteins. Taurine was first isolated about 150 years ago from ox (Taurus) bile. Although taurine can be produced in vivo from cysteine, with the enzymatic help of cysteine dioxygenase, it is mainly acquired from dietary sources, such as meat, eggs, and seafood. The mention of Taurine in the discussion session is determined by the fact that the only two clinical trials found, within our meta-analysis, in the above-mentioned searching platforms, address as primary pathologic condition DM, and H_2_S as the intervention, and these studies indicate Taurine as a drug used.

A wide range of interventions can extend the lifespan and healthspan of H_2_S, including dietary restriction. This is done through the removal of certain nutrients, such as amino acids. One of the most common molecular factors that can affect the longevity of people is the altered metabolism of certain amino acids, such as methionine and cysteine, and the increased production capacity of H_2_S [142]. It is also believed that the presence of H_2_S can delay the onset of aging by blocking the activation of the silent information regulator of the transcription 1 protein (SIRT1). In some studies, it is suggested that the use of dietary restriction for a specific duration can increase the production of H_2_S in rats [143].

As pointed out in the meta-analysis, most patients presented associated cardiovascular diseases (637, representing 51.50% of the total number). It must be emphasized that the main interventions proposed in the eligible clinical trials include Taurine (400 patients representing 32.34% of the total number) and sodium thiosulfate (380 patients representing 30.72% of the total number). It must also be underlined that there was no drug used for 387 patients/subjects (representing 31.29% of the total number), but only the measurement of the H_2_S plasma level.

Various analytical methods have been used to determine the concentration of H_2_S in blood and other tissues, such as fluorescent tools, colorimetry, spectrophotometric analysis, headspace gas determination, polarography, and liquid chromatography-mass spectrometry. Yet, different analysis methods have obtained very diverse intervals of H_2_S concentrations. Furthermore, the levels of H_2_S within tissues and plasma are also significantly different, ranging from 15 nM [18]—for instance, in human plasma—to 300 μM in animal tissues [8] in vivo. These high discrepancies—including as regards to sensitivity and specificity items—can be challenging and, at the same time, require both cautiousness in integrating the related data and must be worthy of further minute research in this domain [18].

In very recent studies, it has been shown that H_2_S can provide therapeutic effects in COVID-19 too. In addition, it is well known that DM is one of the comorbidities which dramatically increases the risk for aggravation of SARS-CoV-2 infection. “Cytokine storm” is a dominant paradigm in explaining the pathogenesis of COVID-19, being involved in many signaling pathways H_2_S influences, including the immune system functioning. Therefore, linking DM to H2S and COVID-19 is an interesting and justified quest direction [37,144,145,146,147].

H_2_S: entrance in the organism, its plasma levels, signaling, metabolism, and their regulation, and also its pathogenic roles, represent topics that warrant an enhanced quest focus. At the same time, being a constituent of sapropelic muds, sulfurous mineral waters, and solfatara—natural sanogenic resources used in balneology—another scientific and practical goal is to promote them based on current, thorough evidence acquired through adequate research activities. Although H_2_S biology and medical usefulness have expanded over the last decades, many related issues/hurdles remain to be further explored, explained, and hopefully overcome [17].

## 5. Conclusions

The available data in this field have revealed interesting sulfur-related biological mechanisms, potentially impacting DM pathophysiology and treatment (as considered, too, empirically in older balneological approaches). Hopefully, future studies will clarify many still poorly known and/or debatable aspects of the subject we approached and pave the way to a better turn to good account, including from bench side to bedside, of the interesting and subtle biological properties of H_2_S.

## Figures and Tables

**Figure 1 ijms-23-06720-f001:**
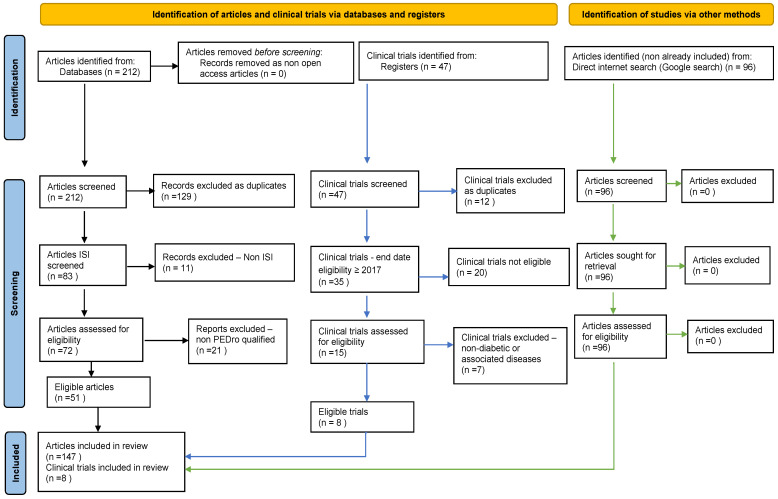
Our adapted PRISMA-type of the flow diagram.

**Figure 2 ijms-23-06720-f002:**
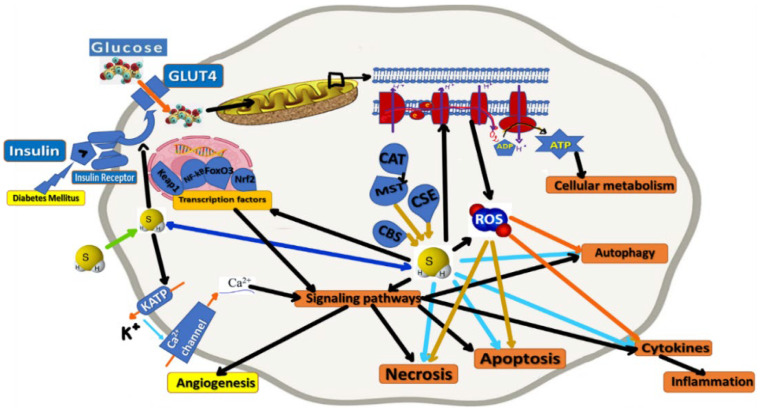
Intimate mechanisms as molecular therapeutic/rehabilitative targets of H_2_S in the case of cells influenced by DM (which impacts glucose uptake, affecting the relation of insulin signal transmission pathway with the cell glucose uptake). Intimate connections are indicated through black arrows, while increasing influences are marked by green arrows and inhibiting or reducing impacts of H2S through blue arrows. Finally, the biosynthesis pathways are stated, represented by cystathionin-β-synthase (CBS), cystathionin-γ-lyase (CSE), and 3-mercaptopyruvate sulfurtransferase (MST), the latter connected with cysteine aminotransferase (CAT).

**Figure 3 ijms-23-06720-f003:**
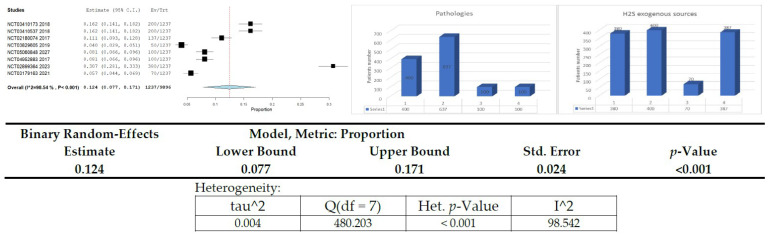
Forest plot related to patients number included in each clinical trial selected for this study.

**Table 1 ijms-23-06720-t001:** The keyword combinations used for the contextual searches in the international databases.

Keywords in Title, Abstract or Author-Specified Keywords	Elsevier	PubMed	PMC	ISI	Total
“Hydrogen sulfide” AND “Diabetes”	8	63	45	59	175
“H2S” AND “Diabetes”	1	14	8	14	37
**Total**	**9**	**77**	**53**	**73**	**212**

**Table 2 ijms-23-06720-t002:** PRISMA resulting conceptual skeleton structure of the article’s organization approach.

*Physiological Properties of H_2_S*		
Authors	Ref. No.	Subject-Data
(Sun, 2021)	[3]	An Updated Insight Into Molecular Mechanism of H_2_S in Cardiomyopathy
(George, 2018)	[6]	Treating inflammation and oxidative stress with H_2_S during age-related macular degeneration
(Zou, 2017)	[10]	H_2_S ameliorates cognitive dysfunction in streptozotocin-induced diabetic rats
(Rey, 2021)	[11]	Mitochondrial metabolism as target of the neuroprotective role of erythropoietin in Parkinson’s disease.
(Testai, 2021)	[12]	Modulation of EndMT by H_2_S in the Prevention of Cardiovascular Fibrosis
(Ciccone, 2021)	[13]	Endothelium as a Source and Target of H_2_S to Improve Its Trophism and Function
(Wu, 2017)	[16]	Exogenous H_2_S facilitating ubiquitin aggregates clearance via autophagy
(Hu, 2017)	[20]	Chelerythrine Attenuates Renal Ischemia/Reperfusion-induced Myocardial Injury
(Kar, 2019)	[22]	H_2_S -mediated regulation of cell death signaling ameliorates adverse cardiac remodeling
(Jeong, 2020)	[24]	Protective effect of H_2_S on oxidative stress-induced neurodegenerative diseases
(Luo, 2019)	[25]	H_2_S upregulates renal AQP-2 protein expression and promotes urine concentration
(Yang, 2019)	[26]	Exogenous H_2_S mitigates myocardial fibrosis through suppression of Wnt pathway
(Liu, 2018)	[27]	H_2_S attenuates myocardial fibrosis through the JAK/STAT signaling pathway
(Sun, 2019)	[28]	Exogenous H_2_S reduces the acetylation levels of mitochondrial respiratory enzymes
(Roa-Coria, 2019)	[29]	Possible involvement of peripheral TRP channels in the H_2_S-induced hyperalgesia
(Yang, 2017)	[30]	Exogenous H_2_S regulates endoplasmic reticulum-mitochondria crosstalk to inhibit apoptosis
(Zhao, 2021)	[31]	H_2_S Plays an Important Role in Diabetic Cardiomyopathy
(Liu, 2017)	[32]	H_2_S modulating mitochondrial morphology to promote mitophagy in endothelial cells
(Qiu, 2018)	[33]	Alpha-lipoic acid regulates the autophagy of vascular smooth muscle cells elevating H_2_S level
(Li, 2017)	[34]	H_2_S reduced renal tissue fibrosis by regulating autophagy in diabetic rats
(Yu, 2020)	[35]	Exogenous H_2_S Induces Hrd1 S-sulfhydration and Prevents CD36 Translocation via VAMP3
(Kar, 2019)	[36]	H_2_S Ameliorates Homocysteine-Induced Cardiac Remodeling and Dysfunction
(Dominic, 2021)	[37]	Decreased availability of nitric oxide and H_2_S is a hallmark of COVID-19
(Loiselle, 2020)	[38]	H_2_S and hepatic lipid metabolism-a critical pairing for liver health
(Ma, 2017)	[39]	Exogenous H_2_S Ameliorates Diabetes-Associated Cognitive Decline
(Jiang, 2020)	[40]	H_2_S Ameliorates Lung Ischemia-Reperfusion Injury Through SIRT1 Signaling Pathway
(Wu, 2019)	[41]	H_2_S Inhibits High Glucose-Induced Neuronal Senescence by Improving Autophagic Flux
** *Pathophysiological Properties H_2_S* **		
**Authors**	**Ref. No.**	**Subject-Data**
(Citi, 2021)	[7]	Role of H2S in endothelial dysfunction: Pathophysiology and therapeutic approaches
(Kang, 2020)	[14]	H_2_S as a Potential Alternative for the Treatment of Myocardial Fibrosis
(Sun, 2019)	[42]	H_2_S and Subsequent Liver Injury
(Szabo, 2017)	[43]	Pharmacological Modulation of H_2_S Levels
(Sun, 2020)	[44]	The Link Between Inflammation and H_2_S
(Zheng, 2020)	[45]	H_2_S protects against diabetes-accelerated atherosclerosis by preventing the activation of NLRP3
(Jia, 2020)	[46]	H_2_S mitigates myocardial inflammation by inhibiting nucleotide-binding oligomerization domain-like receptor protein 3 inflammasome activation in diabetic rats
(Li, 2017)	[47]	H_2_S improves renal fibrosis in STZ-induced diabetic rats by ameliorating TGF-beta 1 expression
(Kar, 20190	[48]	Exercise Training Promotes Cardiac H_2_S Biosynthesis and Mitigates Pyroptosis
(Li, 2019)	[49]	Exogenous H_2_S protects against high glucose-induced apoptosis and oxidative stress
** *H_2_S—Role in Diabetes Mellitus and Associated Vascular Pathology* **		
**Authors**	**Ref. No.**	**Subject-Data**
(Gheibi, 2020)	[8]	Regulation of carbohydrate metabolism by NO and H_2_S: Implications in diabetes
(Zhang, 2021)	[50]	H_2_S regulates insulin secretion and insulin resistance in diabetes mellitus
(Chen, 2021)	[51]	Role of H_2_S in the Endocrine System
(Gheibi, 2019)	[52]	Effects of H_2_S on Carbohydrate Metabolism in Obese Type 2 Diabetic Rats
(Luo, 2017)	[53]	The Role of Exogenous H_2_S in Free Fatty Acids Induced Inflammation in Macrophages
(Comas, 2021)	[54]	The Impact of H_2_S on Obesity-Associated Metabolic Disturbances
(Suzuki, 2017)	[55]	Clinical Implication of Plasma H2S Levels in Japanese Patients with Type 2 Diabetes
(Zhou, 2019)	[56]	H_2_S Prevents Elastin Loss and Attenuates Calcification Induced by High Glucose
** *H_2_S—As a Natural Therapeutic Factor in DM* **		
**Authors**	**Ref. No.**	**Subject-Data**
(Melino, 2019)	[2]	Natural H_2_S Donors from Allium sp. as a Nutraceutical Approach in Type 2 Diabetes
(Sashi, 2019)	[5]	H_2_S inhibits Ca^2+^-induced mitochondrial permeability transition pore opening
(Yang, 2017)	[21]	H_2_S Releasing/Stimulating Reagents
(John, 2017)	[57]	GYY4137, an H_2_S Donor Modulates miR194-Dependent Collagen Realignment
(Bitar, 2018)	[58]	H_2_S Donor NaHS Improves Metabolism and Reduces Muscle Atrophy in Type 2 Diabetes
(Ding, 2017)	[59]	High Glucose Induces Mouse Mesangial Cell Overproliferation via Inhibition of H_2_S Synthesis

**Table 3 ijms-23-06720-t003:** The clinical trials that satisfied all the previous filtering criteria/PRISMA stages selected for qualitative synthesis were included in our meta-analysis to determine the using frequency of the hydrogen sulfide-based interventions for DM or related associated diseases.

No.	Study	Start Year	END Year	N-Total Subjects	Diabetes Mellitus	Cardiovascular/ Associated Disease	Neurodegenerative/ Associated Disease	Respiratory/ Associated Disease	Sodium Thiosulfate	Taurine	Captopril/Enalapril/Hydrochlorothiazide	Observational
**0**	**AXIS LEGEND**				**1**	**2**	**3**	**4**	**1**	**2**	**3**	**4**
**1**	**NCT03410173**	**2017**	**2018**	**200**	**200**					**200**		
**2**	**NCT03410537**	**2017**	**2018**	**200**	**200**					**200**		
**3**	**NCT02180074**	**2013**	**2017**	**137**		**137**						**137**
**4**	**NCT03829605**	**2019**	**2019**	**50**		**50**						**50**
**5**	**NCT05060848**	**2021**	**2027**	**100**			**100**					**100**
**6**	**NCT04952883**	**2016**	**2017**	**100**				**100**				**100**
**7**	**NCT02899364**	**2018**	**2023**	**380**		**380**			**380**			
**8**	**NCT03179163**	**2016**	**2021**	**70**		**70**					**70**	
	**TOTAL**			**1237**	**400**	**637**	**100**	**100**	**380**	**400**	**70**	**387**
	**%**			**100**	**32.34**	**51.50**	**8.08**	**8.08**	**30.72**	**32.34**	**5.66**	**31.29**

## Data Availability

This systematic review was submitted on the PROSPERO-International prospective register of systematic reviews—online platform, No. 293627.

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
