# Peer review of "Recent Advances in Molecular Research on Hydrogen Sulfide (H2S) Role in Diabetes Mellitus (DM)—A Systematic Review"

_ijms, 2022, doi:10.3390/ijms23126720_

Round 1

Reviewer 1 Report

The authors synthesised and systematised the recent literature-related data regarding the therapeutic role of H2S in Diabetes. This systematic review was conducted following the PRISMA methodology. 

This is a well - written and very interesting manuscript which can be published as it is. However, I would encourage the authors to expand the results and discussion on diabetic complications such as nephropathy, retinopathy, and cardiovascular diseases. 

Author Response

Dear reviewer, the authors warmly thank you for your highly professional and detailed analysis of our article. Regarding the suggestion to extend results and discussion with diabetic complications such as nephropathy, retinopathy, and cardiovascular diseases, we take, rather, in considering this subject for a future article, as needing of avoiding information overload. 

We had added the following paragraph: 

In recent studies, H2S was shown to be involved in the regulation of various vascular conditions, such as nephropathy, retinopathy, and neuropathy. H2S-releasing agents could potentially be used as a treatment for diabetes-related endothelial dysfunction. They could help restore the function of the vascular endothelial cells (7) - lines 594-597, citing the article: 

Citi V, Martelli A, Gorica E, Brogi S, Testai L, Calderone V. Role of hydrogen sulfide in endothelial dysfunction: Pathophysiology and therapeutic approaches. J Adv Res [Internet]. 2021;27:99–113. Available from:  https://doi.org/10.1016/j.jare.2020.05.015

In the very recent (2020) cited article we can find useful information regarding your suggested topic, and difficult also to avoid similarity overlap.

Reviewer 2 Report

In the article by Constantin Munteanu et al. the current knowledge of molecular research on hydrogen sulfide (H2S) role in diabetes mellitus (DM) is summarized. Already in the introduction, the authors emphasize that despite extensive research on the participation of H2S in multiple molecular mechanisms, including receptors, membrane ion channels, signaling molecules, enzymes, and transcription factors, the role of this compound in the pathogenesis of diabetes and various angipathies is still unclear. In the introduction, the authors describe diabetes, focusing on its complications. They then characterize the H2S molecule and its role in various diseases. The main value of the work, however, is an extensive literature review. The authors analyzed more than 212 articles describing the role of H2S in various biochemical pathways. The collected knowledge is perfectly systematized and illustrated with clear figures. The meta-analysis conducted by the Authors allows to track how the knowledge and interest in the participation of the described compound in the pathogenesis of many diseases has changed. The article perfectly summarizes the available knowledge.

Author Response

Dear reviewer, the authors warmly thank you for your highly professional and detailed analysis of our article. We express our gratitude for the positive appreciation and constructive comments, pointing out the main achievements of our paper. 

Reviewer 3 Report

This is a very comprehensive review of H2S and glucose metabolism. The diabetes field would appreciate the topic and the detailed review. The review could have been organized better for easier to read and with more clarity—otherwise, exciting area of research and currently a hot topic with caloric restriction-related alterations in H2S. The authors might focus a bit more in protein-calorie restriction and its effects on H2S metabolism.

Author Response

Dear reviewer, the authors warmly thank you for your highly professional and detailed analysis of our article. We express our gratitude for the positive appreciation and constructive comments, pointing out the main achievements of our paper. Regarding caloric restriction-related alterations in H2S - ”The authors might focus a bit more in protein-calorie restriction and its effects on H2S metabolism.”  - we consider this as an appropriate topic for a future article.

We had added in the Discussion section the following paragraph:

A wide range of interventions can extend the lifespan and healthspan, including dietary restriction. This is done through the removal of certain nutrients, such as amino acids. One of the most common molecular factors that can affect the longevity of people is the altered metabolism of certain amino acids, such as methionine and cysteine, and the increased production capacity of H2S (139). It is also believed that the presence of H2S can delay the onset of aging by blocking the activation of the silent information regulator of the transcription 1 protein (SIRT1). In studies, it is suggested that the use of dietary restriction for a specific duration can increase the production of H2S in rats (140).